# OpenReview forum: "Drawing the Line: Enhancing  Trustworthiness of MLLMs Through the Power of Refusal"
_ICLR.cc/2025/Conference — ICLR 2025 Conference Withdrawn Submission_

### Official Review · Reviewer_9ssd · 2024-10-24

**Soundness:** 3
**Presentation:** 2
**Contribution:** 3
**Rating:** 6
**Confidence:** 3

**Summary:**

The authors of this paper aim to address a key issue in multimodal large language models (MLLMs). While MLLMs are powerful and can handle various types of data, they often generate inaccurate or incorrect responses, which diminishes user trust. Many existing approaches overlook the importance of "refusal to answer." In fact, enabling models to refuse answering when information is insufficient can make them more reliable. To address this gap, the authors propose a new framework called InBoL, which trains models to refuse to answer when uncertain. This framework also defines clear criteria for when a refusal is appropriate and introduces a comprehensive data generation process and training strategy to enhance the model's ability to refuse when necessary.

They also present evaluation methods for assessing the model's trustworthiness, focusing on user experience and relevant scoring metrics. Experimental results show that InBoL significantly improves refusal accuracy without compromising the model's usefulness, making it more trustworthy overall.

**Strengths:**

This paper introduces a new data construction pipeline that systematically classifies questions based on model dependence, generates unanswerable questions and organizes them in a standardized format for further training. The confidence estimation method improves the accuracy of the answer by combining string matching and LLM evaluation, ensuring a more robust assessment of model knowledge. The generation of unanswered questions allows the model to learn not to provide incorrect responses when there is insufficient information, but to choose to refuse to answer, thereby improving the credibility of the model and reducing the risk of misleading users. The Advanced training strategies improve model reliability by balancing accuracy and rejection rates, further reducing false information while improving performance. Experiments on different datasets prove the generality and reliability of the method, making this method very valuable for future multimodal research.

**Weaknesses:**

The paper lacks a thorough explanation of how to balance between increasing the refusal rate and maintaining accuracy. The trade-off mechanism between these two aspects is not sufficiently discussed. Additionally, while the current setup may not consider user frustration as a major concern, I’m still curious whether this could become an issue. Would it be beneficial for the model to offer users some form of feedback when refusing to answer, such as providing partial information or clarifying why it cannot respond? This might help improve user experience and prevent frustration by giving more context to the refusal, rather than leaving users without any explanation.

The approach of generating unanswerable questions by randomly swapping images and questions may not effectively simulate real-world unanswerable scenarios. It raise a concern about whether this method accurately reflects realistic situations where a model cannot provide a correct answer.

The paper also falls short in providing examples of unanswerable questions with varying types and levels of difficulty, limiting the demonstration of question diversity. It is unclear whether different prompts or variations in prompt structure are used to encourage this diversity. Furthermore, the strategy for generating unanswerable questions might need to be dynamically adjusted based on the capabilities of different models, but this aspect is not explored.

**Questions:**

Could you elaborate on how you balance increasing the refusal rate with maintaining accuracy? It would be helpful to understand more about the trade-off mechanism between these two goals. Additionally, do you have any strategies in place to address potential user frustration if the model refuses too often?

Do you think the strategy for generating unanswerable questions should be adjusted dynamically depending on the model's capabilities?

---

> ### Author Response · Authors · 2024-11-21
> **Author Response to Reviewer 9ssd(Part 1)**
>
> Thank you for your comprehensive review and insightful comments.We responded in detail as follows:
> > Q1: Could you elaborate on how you balance increasing the refusal rate with maintaining accuracy? It would be helpful to understand more about the trade-off mechanism between these two goals. Additionally, do you have any strategies in place to address potential user frustration if the model refuses too often?
>
> A1: Thank you for your thoughtful question. Balancing the increase in the refusal rate with maintaining accuracy is a crucial aspect of our framework. To address this trade-off, we primarily focus on carefully adjusting the proportions of training data within and beyond the defined boundaries. Detailed experiments can be found in Appendix G.2 of the revised paper.
>
> We also appreciate your point regarding potential user frustration if the model refuses too frequently. While this concern is valid, we believe the benefits of reducing misinformation outweigh the risks of occasional user dissatisfaction. Moreover, we agree that refusal responses accompanied by clear explanation could significantly improve user experience. By explaining why the model cannot respond—such as limitations in visual input or knowledge—users are more likely to perceive the refusal as a thoughtful decision rather than a limitation of the model.
>
> However, implementing this feature is outside the scope of our current work due to the significant increase in complexity it would introduce to both data generation and evaluation processes. Ensuring that these explanations are accurate, relevant, and user-friendly would require substantial additional work. Nevertheless, we see this as a promising direction for future research and plan to explore explanatory feedback mechanisms to make refusal responses more transparent and user-centered. We have added a discussion on this limitation in the Appendix B.
>
> > Q2: The approach of generating unanswerable questions by randomly swapping images and questions may not effectively simulate real-world unanswerable scenarios. It raise a concern about whether this method accurately reflects realistic situations where a model cannot provide a correct answer.
>
> A2: The approach of randomly swapping images and questions is designed to simulate scenarios where the question is unrelated to the image. Such situations are common in real-world contexts, for instance, when a user accidentally uploads the wrong image or asks an unrelated question. Sometimes, users may also intentionally create such mismatches to test the model’s robustness. Furthermore,  similar cases are observed in widely used crowd-sourced datasets like VQAV2, where mismatches between images and questions occur. Therefore, we believe that this method effectively captures certain real-world scenarios.

---

> ### Author Response · Authors · 2024-11-21
> **Author Response to Reviewer 9ssd(Part 2)**
>
> > Q3: The paper also falls short in providing examples of unanswerable questions with varying types and levels of difficulty, limiting the demonstration of question diversity. It is unclear whether different prompts or variations in prompt structure are used to encourage this diversity. Furthermore, the strategy for generating unanswerable questions might need to be dynamically adjusted based on the capabilities of different models, but this aspect is not explored. Do you think the strategy for generating unanswerable questions should be adjusted dynamically depending on the model's capabilities?
>
> A3: Thank you for your insightful feedback. We acknowledge the omission of specific examples of unanswerable questions in the original manuscript. In our revised version, we have included various unanswerable questions in the Appendix D.2 to provide better clarity. Furthermore, Figure 8 in the appendix showcases the prompts used to generate unanswerable questions. These prompts encourage the generation of image-related but unanswerable questions by leveraging GPT-4o, ensuring the diversity in the generated questions.
>
> Regarding the generalizability and robustness of our data generation method, we conducted additional experiments using other advanced open-source MLLMs, such as Qwen2-VL 72B. The unanswerable questions generated by Qwen2-VL 72B yielded results comparable to those produced using GPT-4o, demonstrating the generalizability of our pipeline. Detailed results are provided in Appendix G.3 of the revision of our paper, and an excerpt of the evaluation is shown below:
>
> | Model for data gen | method | Vizwiz(ua) | VQAv2-IDK(filter) | BeyondVisQA |
> |--------------------|--------|------------|-------------------|-------------|
> | GPT-4o             | IDK-IT | 76.01      | 81.42             | 75.25       |
> | GPT-4o             | CA-DPO | 69.97      | 70.63             | 67.75       |
> | Qwen2-VL           | IDK-IT | 74.49      | 79.25             | 72.50       |
> | Qwen2-VL           | CA-DPO | 71.39      | 75.22             | 69.50       |
>
> We also agree with your suggestion that the strategy for generating unanswerable questions could be dynamically adapted based on the capabilities of the models. While our current approach leverages powerful models such as GPT-4o and Qwen2-VL-72B, we acknowledge that weaker or smaller models, such as LLaVA-Next 7B, may require more tailored and sophisticated prompts to generate high-quality unanswerable data. However, given the availability and reliability of existing advanced MLLMs, we have prioritized utilizing these resources for our pipeline, as they effectively minimize the need for such adjustments.

---

> ### Author Response · Authors · 2024-11-25
> **Welcome to Discuss**
>
> Thank you for your insightful feedback and the time spent reviewing our paper. We recognize the importance of comprehensively addressing your concerns and are committed to resolving the issues. Should there be any aspects of our response that you find unclear or if further discussion is needed, please feel free to let us know. We are fully prepared to offer additional clarifications or engage in more detailed discussions to resolve the concerns.
>
> We truly value your meticulous attention and thoughtful insights throughout this process. Your expertise and guidance are indispensable to us, and we eagerly await your further feedback.

---

> > ### Comment · Reviewer_9ssd · 2024-11-28
> >
> > Thank you for your response. I am inclined to maintain a positive evaluation.

---

### Official Review · Reviewer_f1e7 · 2024-11-03

**Soundness:** 3
**Presentation:** 3
**Contribution:** 3
**Rating:** 5
**Confidence:** 3

**Summary:**

This paper presents Information Boundary-aware Learning Framework (InBoL) to enhance the reliability of Multimodal Large Language Models (MLLMs) by systematically training them to recognize their knowledge and perception 'intrinsic' and 'extrinsic' boundaries and refuse to respond when they lack sufficient information. The InBoL framework includes a data generation pipeline to generate structured data (with refusal questions generated with gpt-4o prompting) for 'IDK' Instruction Tuning (IDK-IT) and Confidence-aware Direct Preference Optimization (CA-DPO) (built on top of https://github.com/opendatalab/HA-DPO) — the dataset is designed to improve the model’s accuracy in refusal responses without sacrificing helpfulness to some extent. The paper adopts a user-centric evaluation approach, emphasizing human preference as the core metric for assessing trustworthiness.

**Strengths:**

- the paper disentangles existing value alignment methods (as noted in section 2.1) to create a model-agnostic objective by using user preferences for DPO and generating structured datasets using confidence sampling and LLM judges for creating model fine-tuning recipes.

-  the paper integrates refusal as an explicit mechanism for trustworthiness. This systematic refusal approach seems to be unique among MLLM alignment techniques.

- The paper’s combined approach for instruction tuning for cautious response handling and CA-DPO for adaptive preference optimization proves effective in experimental results, especially for out-of-domain (OOD) tasks.

**Weaknesses:**

- While IDK-IT effectively reduces misinformation, it can limit model's helpfulness (contrary to what the authors claim). One can increase truthfulness by giving refusal responses along with some justification for why the model refused to answer. Authors have not addressed this aspect of refusal with reasoning in the paper.

- For section 3.1, authors give an example for extrinsic information boundary - similarly, it will help to give an example for intrinsic information boundary. Also, Figure 1 examples ii and iii, how can we use the protocol mentioned in the paper to respond appropriately to the questions. The demarkation of intrinsic and extrinsic responses is still confusing.

**Questions:**

- In Table 3, it's not clear if the CA-DPO results also include the IDK-IT step.
- While the paper builds up on intrinsic and extrinsic knowledge sources and that knowledge can come from model's parameters or from the visual content, there are no experiments to dissect this aspect of knowledge grounding. This analysis can significantly improve the quality of the paper (given the framing of the method for increasing trustworthiness)
- There is no comparison with other methods mentioned in the paper for increasing trustworthiness of reducing hallucinations as well as response refusal.

---

> ### Author Response · Authors · 2024-11-22
> **Author Response to Reviewer f1e7(Part 1)**
>
> Thanks for your time and insightful reviews. We responded in detail as follows:
>
> > W1: While IDK-IT effectively reduces misinformation, it can limit model's helpfulness (contrary to what the authors claim). One can increase truthfulness by giving refusal responses along with some justification for why the model refused to answer. Authors have not addressed this aspect of refusal with reasoning in the paper.
>
> A: We appreciate your insightful observation regarding the potential trade-off between IDK-IT's ability to reduce misinformation and its impact on the model’s helpfulness. We acknowledge this limitation in line 414 of the paper.
>
> We agree that providing reasoning alongside refusals could enhance the user experience by clarifying why the model is unable to answer certain queries. This approach would not only improve transparency but also reinforce the model’s self-awareness of its limitations. In fact, we initially considered incorporating explanations into refusal responses. However, we recognized that implementing this feature would considerably increase the complexity of both data generation and evaluation. Adding reasoning to refusals would require extensive additional work to ensure the feedback is accurate, relevant, and user-friendly, which goes beyond the current scope of our work. Nevertheless, we see great potential in this direction and plan to explore explanatory feedback in future work to make refusal responses more informative and user-centered. We have included a discussion of this limitation in Appendix B of the revised version of the paper.
>
> > W2: For section 3.1, authors give an example for extrinsic information boundary - similarly, it will help to give an example for intrinsic information boundary. Also, Figure 1 examples ii and iii, how can we use the protocol mentioned in the paper to respond appropriately to the questions. The demarkation of intrinsic and extrinsic responses is still confusing.
>
> A: We apologize for the confusion. In Figure 1, example iii represents a question that falls outside the extrinsic boundary. For such questions, the model cannot derive the required information from the provided image alone. For instance, if a question asks about a person’s background information that is not visible or implied in the image, the model should refuse to answer. On the other hand,  example ii represents a question that falls within the extrinsic boundary but lies outside the intrinsic boundary. In this case, the image provides enough information to answer the question, but the model needs specific knowledge to interpret it correctly. For example, identifying a specific dog breed in an image requires the model to possess knowledge of dog breeds. If the model lacks this expertise, it should recognize its limitations and refuse to answer, even though the visual information is technically available. In summary, the extrinsic boundary pertains to information that is absent in the input image, whereas the intrinsic boundary relates to the model's own capability limitations in interpreting available information.
>
> > Q1: In Table 3, it's not clear if the CA-DPO results also include the IDK-IT step.
>
> A: Thank you for pointing this out. All CA-DPO results presented in Tables 1, 2, 3, and 4 are based on models that were initially trained with IDK-IT. This detail is discussed in Appendix D, and we apologize for not explicitly mentioning this in the main text.

---

> ### Author Response · Authors · 2024-11-22
> **Author Response to Reviewer f1e7(Part 2)**
>
> > Q2: While the paper builds up on intrinsic and extrinsic knowledge sources and that knowledge can come from model's parameters or from the visual content, there are no experiments to dissect this aspect of knowledge grounding. This analysis can significantly improve the quality of the paper (given the framing of the method for increasing trustworthiness)
>
> A: Thank you for your thoughtful question. Analyzing the distinction between intrinsic and extrinsic information is indeed crucial.
>
> Visual questions inherently rely on both external visual inputs and internal knowledge for accurate responses. To address this, we categorize "unknown" questions into two types based on whether the model fails to answer due to a lack of visual information or internal knowledge. A key question is whether the model can differentiate between these two cases—that is, whether it can identify the specific type of information it lacks.
>
> Therefore, we conducted a linear probing experiment to investigate whether the model’s internal representations encode features that differentiate between these two types of "unknown" cases. We selected 2K "unknown" questions from the training set (with confidence scores below 2) and fed them into the MLLM, extracting the final hidden state of the model's last layer. We then trained a three-layer linear classifier to distinguish between the two types of "unknown" questions. For evaluation, we used a set of 200 "unknown" questions (confidence scores below 2) from the in-domain test set. The classification results are as follows:
>
> | Model | Accuracy (%) |
> | --- | --- |
> | LLaVA 1.5-7B + IDK-IT | 87.01 |
> | LLaVA 1.5-7B + CA-DPO | 80.83 |
>
> Our results demonstrate that linear probing achieves high classification accuracy, indicating that the model’s hidden states effectively encode the distinction between these two types of "unknown" questions. This suggests that the model possesses the potential to recognize the source of missing information, which could be further leveraged to enhance trustworthiness in future work .We have also included this discussion in the revised version of the paper under Appendix G.4, marked in blue for clarity.
>
> > Q3: There is no comparison with other methods mentioned in the paper for increasing trustworthiness of reducing hallucinations as well as response refusal.
>
> A: As noted in line 61 of the paper, the majority of existing approaches for training models to refuse responses are limited to unimodal LLMs and cannot be directly applied to MLLMs. In addition, as for MLLMs, only a few works have explored constructing unanswerable questions to mitigate hallucinations. However, these methods either lack open-sourced model weights [1] or rely on earlier, underperforming MLLMs [2], which prevents a meaningful or fair comparison with our approach.  Fundamentally, these methods focus only on introducing questions beyond the extrinsic boundary and employ supervised fine-tuning (SFT) to train the model to refuse. As stated in lines 404–407 of the paper, we also implemented this approach and presented the results in Table 1 and Table 2 under the "SFT" row. The findings clearly demonstrate that our methods, IDK-IT and CA-DPO, significantly outperform SFT, providing strong evidence of the effectiveness of our proposed framework.
>
> [1] Sungguk Cha, Jusung Lee, Younghyun Lee, and Cheoljong Yang. Visually dehallucinative instruction generation. In ICASSP 2024-2024 IEEE International Conference on Acoustics, Speech and Signal Processing (ICASSP), pp. 5510–5514. IEEE, 2024.
>
> [2] Fuxiao Liu, Kevin Lin, Linjie Li, Jianfeng Wang, Yaser Yacoob, and Lijuan Wang. Mitigating hallucination in large multi-modal models via robust instruction tuning. In The Twelfth International Conference on Learning Representations, 2023b.
>
>
> We hope this clarifies your concerns, and we are grateful for your suggestions.

---

> ### Author Response · Authors · 2024-11-25
> **Welcome to Discuss**
>
> Thank you for your insightful feedback and the time spent reviewing our paper. We recognize the importance of comprehensively addressing your concerns and are committed to resolving the issues. Should there be any aspects of our response that you find unclear or if further discussion is needed, please feel free to let us know. We are fully prepared to offer additional clarifications or engage in more detailed discussions to resolve the concerns.
>
> We truly value your meticulous attention and thoughtful insights throughout this process. Your expertise and guidance are indispensable to us, and we eagerly await your further feedback.

---

> ### Author Response · Authors · 2024-11-28
>
> Dear Reviewer,
>
> We sincerely appreciate your insightful feedback, which has significantly contributed to improving our paper. This is a gentle reminder since we have only a few days until the discussion period ends. If you feel our response and revisions have addressed your concerns, we would be grateful for your continued strong support. Please let us know if you have any additional suggestions for improvement.
>
> Best regards,
>
> The Authors

---

> ### Author Response · Authors · 2024-12-03
>
> Dear Reviewer,
>
> We sincerely appreciate your insightful comments and valuable suggestions on improving our work. As the final deadline approaches, we would kindly like to ask whether our responses have sufficiently addressed your questions and concerns. We are happy to engage in further discussion or provide any clarifications needed, and we welcome any additional feedback to strengthen our work before the rebuttal period ends.﻿ Please kindly let us know if you have additional questions or concerns that stand between us and a higher score! Thank you for your time and thoughtful reviews. ﻿
>
> Best regards,
>
> The Authors

---

### Official Review · Reviewer_sbyo · 2024-11-03

**Soundness:** 3
**Presentation:** 2
**Contribution:** 2
**Rating:** 5
**Confidence:** 4

**Summary:**

This paper presents the Information Boundary-aware Learning Framework (InBoL) for enhancing the trustworthiness of multimodal large language models (MLLMs). MLLMs often produce hallucinated or inaccurate responses, especially when faced with ambiguous or unfamiliar inputs. InBoL addresses this by training models to recognize “information boundaries”—distinguishing between questions they can answer confidently and those they should refuse. The framework leverages two novel training techniques: “I Don’t Know” (IDK) Instruction Tuning (IDK-IT) and Confidence-aware Direct Preference Optimization (CA-DPO), both aimed at improving refusal responses for uncertain or ambiguous queries.

To evaluate trustworthiness, the authors introduce a user-centered metric that rewards accurate and helpful responses while penalizing misinformation. Experimental results indicate that InBoL improves refusal accuracy without compromising the helpfulness of responses, setting a new benchmark for trustworthiness training in MLLMs. This work proposes a robust approach to model alignment for safe and reliable AI responses, particularly in vision-language tasks.

**Strengths:**

The paper leverages a decrease in accuracy robustness of some VLLM’s responses to create a dataset that establishes clear boundaries for confident responses, incorporating refusal mechanisms for cases where the model might otherwise respond incorrectly. Through training, combined with experimentation, the study shoes the reduction of unreliable responses through the use of synthetic data in training.

**Weaknesses:**

- Limited Scope: Experiments focus narrowly on Visual LLMs (with claim for MLLMs) and small models (e.g., LLaVA 1.5), lacking insights into broader MLLM applicability or resource requirements.

- Lack of Explanatory Rejection: The model is trained to learn outright refusals lack explanations. Adding explanations or visualizations of boundaries could enhance user trust.

- Conceptual and Metric Issues: The construction of the "unknown" dataset used for training lacks clear justification for how it genuinely promotes trustworthiness in LLMs. Furthermore, the trustworthiness score proposed in the paper is unnormalized, which can be different across different dataset with different theoretical max accuracy rate. Answered accuracy might be a more objective measure. There’s also a lack of motivation behind the metrics, authors say the old metric is bad because “unknown” questions domain is hard to know, but then create such domain in Fig 3)

- Boundary and Hallucination Exploration: The simplistic boundary approach doesn’t adequately address the cause of hallucination, or accounting for decrease accuracy of responses. This investigation of cause of not refusal is lacking.

- Methodology Novelty: Core techniques (PEFT, DPO) are standard, limiting innovation, and newer methods like inference-time corrections could improve boundary handling.

- Dependence on GPT-4o: Dependence on GPT-4o for data generation risks biases from its potential hallucinations. Limited discussion of pipeline generalizability weakens applicability.

- Missing Literature, Limitations and Ethics Review: A more complete literature review and ethical discussion are needed (not in the appendix), currently relegated to the appendix, to frame the work’s limitations and ethical impact fully. Also the cost of InBol is lacking, which can be heavy to generate such dataset with model responses of so many benchmark in practice.

**Questions:**

- How does InBoL innovate beyond existing methods like PEFT and DPO? While InBoL uses “I Don’t Know” (IDK) Instruction Tuning and Confidence-aware Direct Preference Optimization (CA-DPO), which build on established techniques, could you clarify any additional features that make InBoL uniquely suited to MLLMs? Consider highlighting or incorporating advanced techniques, such as inference-time adjustments, to distinguish InBoL further.
- Why did you choose an unnormalized trustworthiness score over a normalized or cross-comparable metric? The score is dataset-dependent, which could affect its generalizability across datasets. Could you discuss why this metric was prioritized over an alternative like answered accuracy? Introducing a normalized metric or justifying this choice could clarify how trustworthiness is assessed across various MLLMs and test sets.
- How does InBoL’s boundary creation address hallucination causes and refusal failures? Defining boundaries by confidence thresholds doesn’t appear to directly address why hallucinations or refusal failures might occur. Would expanding boundary classifications or including visual boundary explanations aid in understanding these limitations?
- How does the dependence on GPT-4o for data generation impact generalizability? Relying on GPT-4o to generate unanswerable questions may risk inheriting its biases. Could you describe any mitigations for GPT-4o-induced biases in the dataset and discuss InBoL’s performance if an alternative model or method is used to create unanswerable data?
- What are your expectations for InBoL’s generalizability to larger models and domains beyond visual tasks? The experiments focus on smaller models (e.g., LLaVA 1.5) and visual LLMs. Providing experimental insights on InBoL’s scalability to more complex or varied models would clarify its broader applicability.
- Would explanatory feedback enhance trust in refusal responses? InBoL’s refusal responses currently lack explanations, potentially limiting user trust. Have you considered integrating explanatory refusals to improve user understanding, particularly when refusals stem from visual limitations or knowledge boundaries?
- Could you expand on the ethical implications of refusal mechanisms? The ethical considerations section is limited, mainly relegated to the appendix. Given the potential for user reliance on refusal responses, would a more thorough discussion on the ethical impacts of this feature strengthen the paper’s contextual relevance?

---

> ### Author Response · Authors · 2024-11-21
> **Author Response to Reviewer sbyo(Part 1)**
>
> Thank you for your review and valuable feedback on our paper. We responded in detail as follows:
>
> > (Q1 & W5) Methodology Novelty: How does InBoL innovate beyond existing methods like PEFT and DPO? While InBoL uses “I Don’t Know” (IDK) Instruction Tuning and Confidence-aware Direct Preference Optimization (CA-DPO), which build on established techniques, could you clarify any additional features that make InBoL uniquely suited to MLLMs? Consider highlighting or incorporating advanced techniques, such as inference-time adjustments, to distinguish InBoL further.
>
> A1: Thank you for your comments. The InBoL framework is distinct in its explicit incorporation of refusal as a mechanism to enhance trustworthiness, training MLLMs to recognize and respond appropriately when they lack sufficient information, which is  unique among MLLM alignment techniques. To achieve this, InBoL introduces the information boundaries (Section 3.1) and includes a comprehensive data construction pipeline (Section 3.2) specifically designed to generate high-quality training data for refusal responses. Therefore, the core innovations of our framework include not only IDK-IT and CA-DPO but also the introduction of information boundary and a detailed  construction pipeline of multimodal instruction with refusal response.
> While we recognize that inference-time adjustments could potentially enhance boundary handling further, our primary focus is on training-based methods to instill boundary awareness in MLLMs, leaving inference-time optimizations as a promising direction for future exploration.
>
> > (Q2 & W3)Conceptual and Metric Issues: Why did you choose an unnormalized trustworthiness score over a normalized or cross-comparable metric? The score is dataset-dependent, which could affect its generalizability across datasets. Could you discuss why this metric was prioritized over an alternative like answered accuracy? Introducing a normalized metric or justifying this choice could clarify how trustworthiness is assessed across various MLLMs and test sets.
>
> A2: The motivation behind our proposed evaluation method stems from limitations observed in prior approaches, which often require constructing "unknown" test set for each model. This construction process incurs high computational costs due to the need for multiple sampling. For model training, these sampling costs are manageable, allowing us to collect "unknown" questions as described in Figure 3; however, for evaluation purposes, such a costly approach would limit practicality across models.
>
> The choice of metrics is fundamentally tied to the definition of trustworthiness. In this work, we adopt a user-centric perspective, proposing that a trustworthy model should maximize helpful responses while minimizing misinformation. Accordingly, we define a user-centered value function and further propose the trustworthiness score that reflects a balanced view of trustworthiness by taking both accuracy and refusal rate into account.
>
> While "Answered Accuracy" might be considered as an alternative, it has a significant limitation: it tends to encourage overly conservative behavior at the expense of overall accuracy.If Answered Accuracy were used as the primary metric, the optimal strategy for a model would be to refuse all questions where its confidence is less than 1. While this could lead to an ideal Answered Accuracy close to 100%, it would severely compromise the model’s helpfulness by refusing many questions where it could have provided useful and accurate responses.
>
> In contrast, our trustworthiness score strikes a better balance. By explicitly assigning a score of 1 to correct responses and 0 to refusal responses, the metric incentivizes models to maintain high accuracy while leveraging refusal as a safeguard against misinformation. This balanced approach encourages models to provide as many correct responses as possible while prudently refusing only when necessary, thereby aligning with our expectations of trustworthiness in MLLMs.
>
> If we have misunderstood your question or if there is any aspect of our response that is unclear, please feel free to let us know.

---

> ### Author Response · Authors · 2024-11-21
> **Author Response to Reviewer sbyo(Part 2)**
>
> > (Q3 & W4) Boundary and Hallucination Exploration: How does InBoL’s boundary creation address hallucination causes and refusal failures? Defining boundaries by confidence thresholds doesn’t appear to directly address why hallucinations or refusal failures might occur. Would expanding boundary classifications or including visual boundary explanations aid in understanding these limitations?
>
> A3: The information boundaries introduced in InBoL are designed to provide a guiding framework for systematically identifying and managing scenarios where the model lacks sufficient information. Based on these identified cases, we developed a data generation pipeline and training strategies (IDK-IT and CA-DPO) to train the model to refuse to answer when adequate information is unavailable, thereby reducing the occurrence of hallucinations. One of the primary causes of hallucinations in MLLMs is their inability to recognize when a query lies beyond their knowledge or perceptual capacity. The InBoL framework specifically addresses this challenge by training the model to appropriately refuse to answer in such situations, thereby directly address the cause of hallucinations.
>
> It is important to highlight that boundary creation is not intended to eliminate refusal failures but rather serves as a guiding framework to help the model learn to refuse appropriately. In fact, refusal failures are inherently unavoidable. First, accurately assessing its own boundaries(confidence) is a challenge for models. Second, as demonstrated in Section 4.3.2, we observe that the model occasionally attempts to answer questions with low confidence, resulting in some refusal failures. However, this behavior ensures a balance between maintaining the model's overall performance and improving its refusal rates.
>
> If we have misunderstood your question or if there is any aspect of our response that is unclear, please feel free to let us know.
>
> > (Q4 & W6) Dependence on GPT-4o: How does the dependence on GPT-4o for data generation impact generalizability? Relying on GPT-4o to generate unanswerable questions may risk inheriting its biases. Could you describe any mitigations for GPT-4o-induced biases in the dataset and discuss InBoL’s performance if an alternative model or method is used to create unanswerable data?
>
> A4: Thank you for raising this important question. First, it is important to note that only a small number of the unanswerable data in our experiments was generated using GPT-4o (approximately 1k samples). As a result, the overall impact of GPT-4o-generated data on the model’s performance is limited.
>
> Second, we carefully designed prompts for GPT-4o to ensure that the generated unanswerable questions were strongly related to the accompanying images. Specifically, GPT-4o was instructed to create unanswerable questions from 2 perspective as shown in Figure 8 in appendix. Additionally, we implemented a filtering mechanism to address potential hallucination issues from GPT-4o, ensuring that only high-quality unanswerable questions were included in the dataset.
>
> Finally, we conducted additional experiments using open-source MLLM(Qwen2-VL-72B-Instruct) to generate and filter data. The training results on this dataset were comparable to those achieved with GPT-4o-generated data, highlighting the generalizability of our pipeline. In the revised version of the paper, we have included a new section in Appendix G.3 to discuss this issue in greater depth, with updates highlighted in blue for clarity. More detailed results can also be found in this section.
>
> | Model for data gen | method | Vizwiz(ua) | VQAv2-IDK(filter) | BeyondVisQA |
> |--------------------|--------|------------|-------------------|-------------|
> | GPT-4o             | IDK-IT | 76.01      | 81.42             | 75.25       |
> | GPT-4o             | CA-DPO | 69.97      | 70.63             | 67.75       |
> | Qwen2-VL           | IDK-IT | 74.49      | 79.25             | 72.50       |
> | Qwen2-VL           | CA-DPO | 71.39      | 75.22             | 69.50       |

---

> ### Author Response · Authors · 2024-11-21
> **Author Response to Reviewer sbyo(Part 3)**
>
> > (Q5 & W1)Limited Scop: What are your expectations for InBoL’s generalizability to larger models and domains beyond visual tasks? The experiments focus on smaller models (e.g., LLaVA 1.5) and visual LLMs. Providing experimental insights on InBoL’s scalability to more complex or varied models would clarify its broader applicability.
>
> A5: Due to computational resource limitations, our experiments primarily focused on LLaVA 1.5 with 7B and 13B parameters. To further evaluate the scalability and generalization of InBoL, we have conducted additional experiments on a more advanced model, LLaVA-Next 7B. These results demonstrate that InBoL effectively enhances trustworthiness for LLaVA-Next 7B, indicating its potential applicability to larger and more advanced models.
>
> | Method          | ID-Overall Acc | ID-Overall RefR | ID-Overall Score | AOKVQA Acc | AOKVQA RefR | AOKVQA Score | GQA Acc | GQA RefR | GQA Score | BeyondVisQA RefR | MMMU Acc | MMMU RefR | MMMU Score | MMBench Acc | MMBench RefR | MMBench Score |
> |-----------------|----------------|------------------|------------------|------------|-------------|--------------|---------|----------|-----------|------------------|----------|-----------|------------|-------------|--------------|---------------|
> | Original        | 54.50          | 5.00            | 14.00           | 82.97      | 0.00        | 65.94        | 61.00   | 0.00     | 22.00     | 16.50            | 35.80    | 0.00      | -28.39     | 63.49       | 0.00         | 26.98         |
> | Refusal Prompt  | 51.90          | 9.40            | 13.20           | 70.41      | 16.38       | 57.20        | 60.47   | 2.12     | 23.06     | 48.00            | 31.78    | 10.44     | -26.00     | 61.25       | 0.34         | 22.84         |
> | SFT             | 53.10          | 12.30           | 18.50           | 75.81      | 5.85        | 57.47        | 61.16   | 1.21     | 23.53     | 76.25            | 35.78    | 1.00      | -27.44     | 64.95       | 0.17         | 30.07         |
> | IDK-IT          | 46.10          | 29.90           | 22.10           | 66.99      | 22.95       | 56.93        | 50.75   | 23.75    | 25.25     | **76.25**            | 19.22    | 55.67     |**-5.89**      | 58.42       | 16.75        | 33.59         |
> | CSA-DPO         | 52.60          | 29.50           | **34.70**           | 76.55      | 14.31       | **67.41**        | 60.50   | 11.75    | **32.75**     | 71.75            | 23.22    | 45.11     | -8.44      | 60.40       | 18.64        | **39.43**         |
>
>
> > (Q6 & W2)Lack of Explanatory Rejection: Would explanatory feedback enhance trust in refusal responses? InBoL’s refusal responses currently lack explanations, potentially limiting user trust. Have you considered integrating explanatory refusals to improve user understanding, particularly when refusals stem from visual limitations or knowledge boundaries?
>
> A6: Thank you for your insightful question. We agree that providing explanatory feedback in refusal responses has the potential to significantly enhance user trust. By offering clear explanations for why a refusal occurs—such as limitations in visual input or knowledge boundaries—the model could improve not only its self-awareness but also the user experience by fostering a greater understanding of its reasoning.
>
> We did consider incorporating explanatory refusals during the development of our framework. However, we found that doing so would substantially increase the complexity of both the data generation and evaluation processes. This additional complexity would require extensive work to ensure that the explanations are accurate,  relevant, and user-friendly. As such, we chose to only focus on basic refusal responses in this study. We recognize this as a limitation of our current work and have included a discussion on it in Appendix B of the revised paper. Exploring explanatory feedback is a promising direction for future research, and we plan to address this in subsequent studies.

---

> ### Author Response · Authors · 2024-11-21
> **Author Response to Reviewer sbyo(Part 4)**
>
> > (Q7 & W7)Missing Literature, Limitations and Ethics Review: Could you expand on the ethical implications of refusal mechanisms? The ethical considerations section is limited, mainly relegated to the appendix. Given the potential for user reliance on refusal responses, would a more thorough discussion on the ethical impacts of this feature strengthen the paper’s contextual relevance?
>
> A7: Thank you for your thoughtful and constructive feedback.**
>
> We greatly appreciate your suggestion to expand on the ethical implications of refusal mechanisms and to provide a more comprehensive discussion on the limitations of our work. Below, we address these points in detail:
>
> 1. Ethical Considerations of Refusal Mechanisms:
> We believe that the refusal mechanism plays a crucial role in reducing users' dependency on the model, thereby minimizing the risk of inadvertently misleading users. As such, we regard this mechanism as ethically sound. However, we acknowledge that refusal mechanisms may have unintended consequences, such as causing user frustration or dissatisfaction. While the core concept remains ethically robust, we recognize the importance of further discussion on its limitations, particularly in the context of user experience.
>
> 2. Limitations:
> In this work, we did not explore the generation of explanations for refusal responses, an important and underexamined area. From the model's perspective, many questions require reasoning processes to determine whether sufficient information is available to provide an accurate answer. By incorporating explanations for refusal responses, the model could better learn when to refuse appropriately, thereby enhancing its awareness of its own limitations and boundaries. From the user's perspective, unexplained refusals may lead to confusion or dissatisfaction. Providing clear and interpretable justifications for refusals could make the refusal mechanism more transparent and user-friendly, significantly improving the overall user experience.
> For future work, we plan to focus on enabling the model to generate well-reasoned and contextually appropriate refusal explanations. This will involve developing methodologies for constructing relevant datasets and designing robust evaluation frameworks to assess the quality and relevance of the generated explanations. By making refusal responses more informative and transparent, we aim to further enhance the trustworthiness of the model while ensuring a more positive and engaging user experience.
> We have added this limitation to the revised version of Appendix B for clarity.
>
> 3. Cost of InBoL Implementation:
> Regarding the computational cost of implementing InBoL, we emphasize that the framework does not require the use of the entire VQAv2, Oven, or SQA datasets. For IDK-IT and CA-DPO, we constructed approximately 11k and 24k samples, respectively. Generating datasets of this scale is computationally manageable and does not incur significant overhead.
>
> We hope this clarifies your concerns, and we are grateful for your suggestions.

---

> ### Author Response · Authors · 2024-11-25
> **Welcome to Discuss**
>
> Thank you for your insightful feedback and the time spent reviewing our paper. We recognize the importance of comprehensively addressing your concerns and are committed to resolving the issues. Should there be any aspects of our response that you find unclear or if further discussion is needed, please feel free to let us know. We are fully prepared to offer additional clarifications or engage in more detailed discussions to resolve the concerns.
>
> We truly value your meticulous attention and thoughtful insights throughout this process. Your expertise and guidance are indispensable to us, and we eagerly await your further feedback.

---

> ### Author Response · Authors · 2024-11-28
>
> Dear Reviewer,
>
> We sincerely appreciate your insightful feedback, which has significantly contributed to improving our paper. This is a gentle reminder since we have only a few days until the discussion period ends. If you feel our response and revisions have addressed your concerns, we would be grateful for your continued strong support. Please let us know if you have any additional suggestions for improvement.
>
> Best regards,
>
> The Authors

---

> ### Author Response · Authors · 2024-12-03
>
> Dear Reviewer,
>
> We sincerely appreciate your insightful comments and valuable suggestions on improving our work. As the final deadline approaches, we would kindly like to ask whether our responses have sufficiently addressed your questions and concerns. We are happy to engage in further discussion or provide any clarifications needed, and we welcome any additional feedback to strengthen our work before the rebuttal period ends.﻿ Please kindly let us know if you have additional questions or concerns that stand between us and a higher score! Thank you for your time and thoughtful reviews. ﻿
>
> Best regards,
>
> The Authors

---

### Note · Authors · 2024-12-07

I have read and agree with the venue's withdrawal policy on behalf of myself and my co-authors.